# Ultra-short pulse laser acceleration of protons to 80 MeV from cryogenic hydrogen jets tailored to near-critical density

Martin Rehwald [1,2] ✉, Stefan Assenbaum [1,2], Constantin Bernert [1,2], Florian-Emanuel Brack [1,2], Michael Bussmann [1,3], Thomas E. Cowan [1,2], Chandra B. Curry [4,5], Frederico Fiuza [4], Marco Garten [1,9], Lennart Gaus [1,2], Maxence Gauthier [4], Sebastian Göde [6], Ilja Göthel [1,2], Siegfried H. Glenzer [4], Lingen Huang [1], Axel Huebl [1,9], Jongjin B. Kim [4], Thomas Kluge [1], Stephan Kraft [1], Florian Kroll [1], Josefine Metzkes-Ng [1], Thomas Miethlinger [1,2], Markus Loeser [1], Lieselotte Obst-Huebl [1,9], Marvin Reimold [1,2], Hans-Peter Schlenvoigt [1], Christopher Schoenwaelder [4,7], Ulrich Schramm [1,2], Mathias Siebold [1], Franziska Treffert [4,8], Long Yang [1,2], Tim Ziegler [1,2] & Karl Zeil [1]

Laser plasma-based particle accelerators attract great interest in fields where conventional accelerators reach limits based on size, cost or beam parameters. Despite the fact that particle in cell simulations have predicted several advantageous ion acceleration schemes, laser accelerators have not yet reached their full potential in producing simultaneous high-radiation doses at high particle energies. The most stringent limitation is the lack of a suitable high-repetition rate target that also provides a high degree of control of the plasma conditions required to access these advanced regimes. Here, we demonstrate that the interaction of petawatt-class laser pulses with a pre-formed micrometer-sized cryogenic hydrogen jet plasma overcomes these limitations enabling tailored density scans from the solid to the underdense regime. Our proof-of-concept experiment demonstrates that the near-critical plasma density profile produces proton energies of up to 80 MeV. Based on hydrodynamic and three-dimensional particle in cell simulations, transition between different acceleration schemes are shown, suggesting enhanced proton acceleration at the relativistic transparency front for the optimal case.

Laser-driven ion accelerators[1,2] have gained relevance in a wide range of multidisciplinary applications[3]. The short, intense, and low emittance[4] ion pulses excel in warm dense matter research[5], archaeological surveys[6], fusion energy research[7,8], time-resolved studies of transient fields[9–11], and research in radiation oncology[12–15]. These applications benefit from unprecedented high dose rates in a single laser shot and further need the development of increased ion energies and improved control over the spectral and spatial ion distributions.

[1]Helmholtz-Zentrum Dresden - Rossendorf, Institute of Radiation Physics, Bautzner Landstr. 400, 01328 Dresden, Germany. [2]Technische Universität Dresden, 01062 Dresden, Germany. [3]Center for Advanced Systems Understanding (CASUS), 02826, Görlitz, Germany. [4]High Energy Density Science Division, SLAC National Accelerator Laboratory, Menlo Park, CA 94025, USA. [5]University of Alberta, Edmonton, Alberta T6G 1H9, Canada. [6]European XFEL GmbH, Holzkoppel 4, 22869 Schenefeld, Germany. [7]Friedrich-Alexander Universität Erlangen-Nürnberg, 91054 Erlangen, Germany. [8]Technische Universität Darmstadt, 64289 Darmstadt, Germany. [9]Present address: Lawrence Berkeley National Laboratory, Berkeley, CA 94720, USA. ✉e-mail: m.rehwald@hzdr.de

Currently, high-power lasers are routinely approaching the petawatt (PW) level at intensities exceeding $10^{21}$ W cm$^{-2}$ [3]; these lasers enable the exploitation of advanced acceleration schemes using specialized targets and tailored temporal laser pulse shapes[16]. Hence, defining the ideal, technically realizable, and robust laser target configuration has been the subject of extensive studies.

When propagating into a dense plasma, light is reflected at the surface of the critical density $n_c = \omega_L^2 m_e/(4\pi e^2)$ with $\omega_L$ the laser frequency, $m_e$ the electron mass and $e$ the electron charge. For intense laser pulses, plasma electrons are accelerated to relativistic energies and thus increase in mass $\gamma m_e$, causing relativistic transparency at densities $n_c^{rel} = \gamma n_c > n_c$. The target density is the decisive parameter for the type of laser-matter interaction and therefore determines the ion acceleration mechanism. This ranges from target-normal sheath acceleration (TNSA)[17–19] and hole-boring or light-sail radiation pressure acceleration (HB-RPA, LS-RPA)[20–23] at solid density over acceleration from targets undergoing relativistic transparency[24–27] and synchronized acceleration at the relativistic critical density surface (RTF-RPA)[28,29] to collisionless shock acceleration (CSA)[30] and magnetic vortex acceleration (MVA)[31,32] at near-critical density. A major advantage of acceleration mechanisms in the near-critical density regime is their promise of higher ion beam energies compared to TNSA. However, while TNSA is thoroughly investigated, experimental identification of the other, often competing mechanisms is a major challenge and mostly relies on dedicated numerical modeling. Typically, multiple mechanisms coexist and transitions may enhance the acceleration performance as illustrated by a transparency-enhanced hybrid TNSA-RPA regime yielding the highest reported energies of over 90 MeV achieved at a high-energy laser system[33].

Key to exploring and optimizing different acceleration schemes is the control of the target density profile and its temporal evolution. This requires simultaneous tailoring of bulk density and surface density gradients[34], ideally using one single target concept and material species to avoid complicated ionization dynamics. So far, density variations have been realized by a multitude of different target systems, such as overdense foils[33,35] or targets exploded by heater pulses[36,37]. Another approach is to manufacture near critical density but still relativistically overdense targets by nanoscopic structuring of different composites (e.g. foams)[38,39] or by directly employing low-density materials like solid hydrogen[40–42]. Underdense systems have been supplied by high-pressure gas jets[43,44]. Yet, full control of the acceleration mechanisms and determining the precise experimental characterization of the plasma conditions, which are needed as input parameters for simulations has been challenging. Limitations arise from technical constraints, e.g., from the target geometry, or the measurements require ultra-fast probing techniques in the X-ray regime to penetrate the otherwise opaque solid density targets[45,46].

Here, we report on proof-of-concept experiments irradiating a self-replenishing, debris-free, cylindrical cryogenic hydrogen jet with PW-class ultra-short laser pulses yielding high energy proton beams of up to 80 MeV. Two-color optical probing allowed for measurements of the spatiotemporal target evolution, providing access to the on-shot target density profile. Low-intensity laser pre-pulses were intentionally introduced to hydrodynamically pre-expand the target before proton acceleration was triggered by the main PW-laser pulse featuring ultra-high temporal contrast[16]. While walking through different acceleration mechanisms realized during the transition between the overdense and the underdense regime, the low solid-density (30 $n_c$) of the hydrogen jets proved to be favorable to support sufficiently high bulk densities with still steep gradients at the solid-vacuum interface. The combination of a well-characterized laser pulse, the on-shot target expansion control and the nature of the hydrogen jet being of low density and single ion species enabled the realization of a start-to-end simulation framework based on hydrodynamic simulations and three-dimensional particle in cell studies. The experimental observation of an increase in maximum proton energy from 40 MeV to 80 MeV with decreasing target density could thus be identified as a transition from TNSA at solid density to enhanced acceleration at the relativistic critical density front that is propagating with the laser into the bulk of the target.

## Results

The experiment was carried out at the Draco PW laser. Ultra-short laser pulses of 18 J energy were focused onto a solid hydrogen jet of 5 $\mu$m diameter[47] yielding intensities of $5.4 \cdot 10^{21}$ W cm$^{-2}$ (Fig. 1). When fully ionized, the target density amounts to $5.1 \cdot 10^{22}$ cm$^{-3}$ or 30 $n_{c,800nm}$ with $n_{c,800nm}$ being the critical density for the laser wavelength of 800 nm. The temporal laser pulse contrast was enhanced by a re-collimating

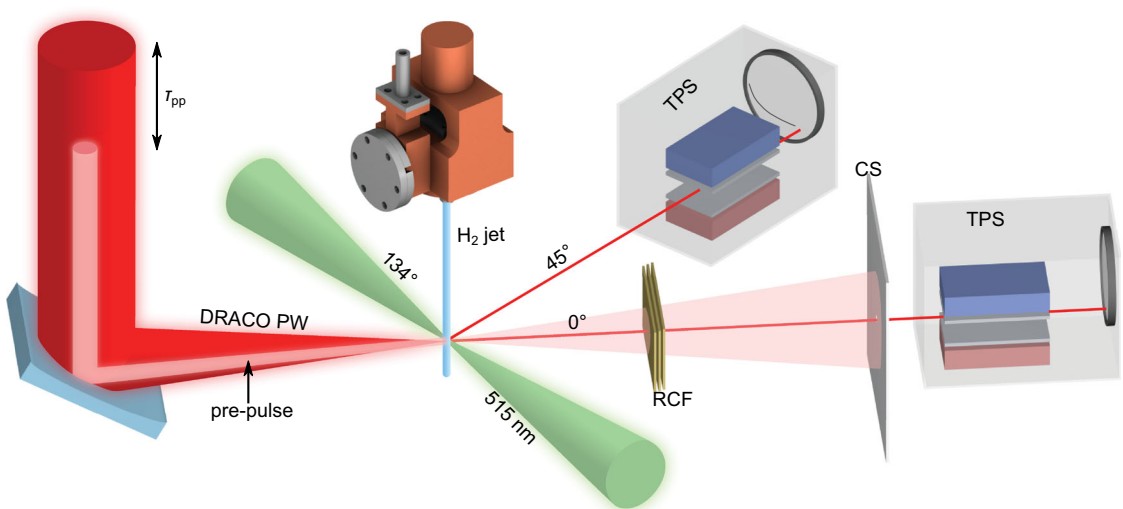

**Fig. 1 | Experimental setup.** DRACO PW laser pulses (red) are focused onto a cylindrical cryogenic hydrogen jet target (blue) pre-expanded using pre-pulses of variable time delay $\tau_{pp}$. A synchronized optical probe laser pulse with 515 nm wavelength (green) was used as a backlighter for shadowgraphy imaging with a spatial resolution of 1.5 $\mu$m. Thomson parabola spectrometers (TPS), as well as optional radiochromic film stacks (RCF) were positioned along the laser propagation direction and along the 45° axis for proton energy spectra measurement. A ceramic screen (CS) captured laser light behind the target. Image credit: Greg Stewart, SLAC National Accelerator Laboratory.

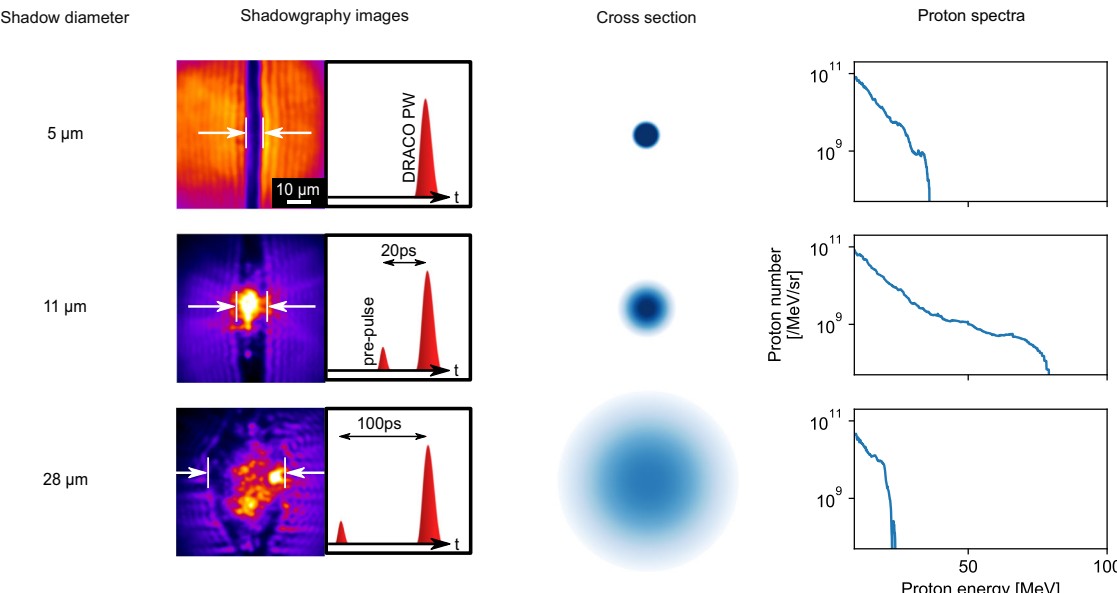

**Fig. 2 | Measurements of proton energy spectrum.** On-shot shadowgraphy images with the adjacent timing scheme, cross sections visualizing the target density profile and proton energy spectra, as measured by the TPS positioned along the laser propagation direction are shown for different jet pre-expansions (shadow diameter are provided in the left for each case and are indicated by the white arrows in the shadowgraphy images). The rows refer to the shots labeled (1)-(3) in Fig. 3a.

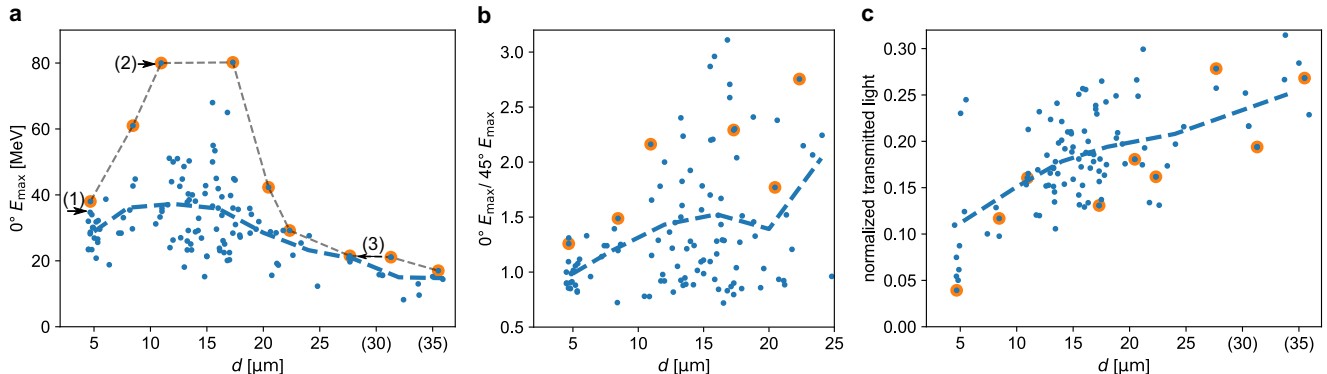

**Fig. 3 | Experimental target density scan showing maximum proton energies, emission distributions, and transmitted light fractions. a** Acquired data set of maximum proton energies $E_{max}$ detected in laser propagation direction with respect to the shadow diameter $d$ determined by optical probing for each shot. The blue dashed line indicates the trend of the average $E_{max}$ as a guide for the eye. The orange circles highlight best performing shots within 4 $\mu m$ bins of the shadow diameter. The labels (1)-(3) refer to the proton energy spectra in Fig. 2 (rows from top to bottom). **b** shows the ratio between $E_{max}$ measured along 0° with respect to the 45° emission direction, increasing with growing pre-expansion of the target. **c** The laser light which is transmitted through the jet, captured by the ceramic screen and normalized to shots without a target increases with growing pre-expansion. Shadow diameter values ≥ 30 $\mu m$ in **a**, **c** are set in brackets as they correspond to targets which are already transparent to the 515 nm probe beam light, and therefore only represent a lower limit.

single plasma mirror[16]. This ensured an unperturbed target until two picoseconds prior to the peak of the pulse[48]. This level of control of laser contrast conditions allows tailoring of the target density profile the main pulse interacts with. For this purpose a low-intensity pre-pulse ($5.8 \cdot 10^{17}$ W cm⁻², 55 fs pulse duration, 32 $\mu m$ × 19 $\mu m$ spot size) with variable delay (10 ps - 170 ps) was introduced to trigger an isotropic hydrodynamic pre-expansion of the target to adjust its core density. For each individual full energy shot, the expansion of the target at the time of the main pulse arrival was confirmed by the measurement of the shadow diameter $d$ using on-shot high-resolution shadowgraphy images (second to left column in Fig. 2). These were recorded employing a synchronized off-harmonic (515 nm wavelength) optical probe laser enabling the collection of unsaturated probe images at full main pulse energy[48–50] with a spatial resolution of 1.5 $\mu m$.

Two Thomson parabola spectrometers characterized the proton emission. The resulting proton spectra (Fig. 2), recorded in laser propagation direction, indicate optimal performance for a shadow diameter of $d = 11 \mu m$ with proton energies of up to 80 MeV. No pre-expansion ($d = 5 \mu m$) or larger pre-expansion ($d = 28 \mu m$) yielded lower maximum energies as well as reduced particle yield. A scan of the maximum proton energy, measured in laser propagation direction, is presented in Fig. 3a as a function of the shadow diameter. The coupling efficiency of laser energy into the target is strongly dependent on the spatial overlap of jet (5 $\mu m$ diameter) and laser focus (3 $\mu m$ FWHM) position (for a detailed discussion refer to Supplementary Fig. 7). As the most accurate criterion accessible, the fraction of the light bypassing the target and recorded using the transmitted light screen was used for coarse selection of central hits (see methods) which leads to the data shown in Fig. 3 (1/3 of the about 350 laser shots conducted during three consecutive days). Despite possible contributions of varying laser pulse parameters, we assume that (as later strengthened by systematic simulations of this behaviour) the remaining scatter of

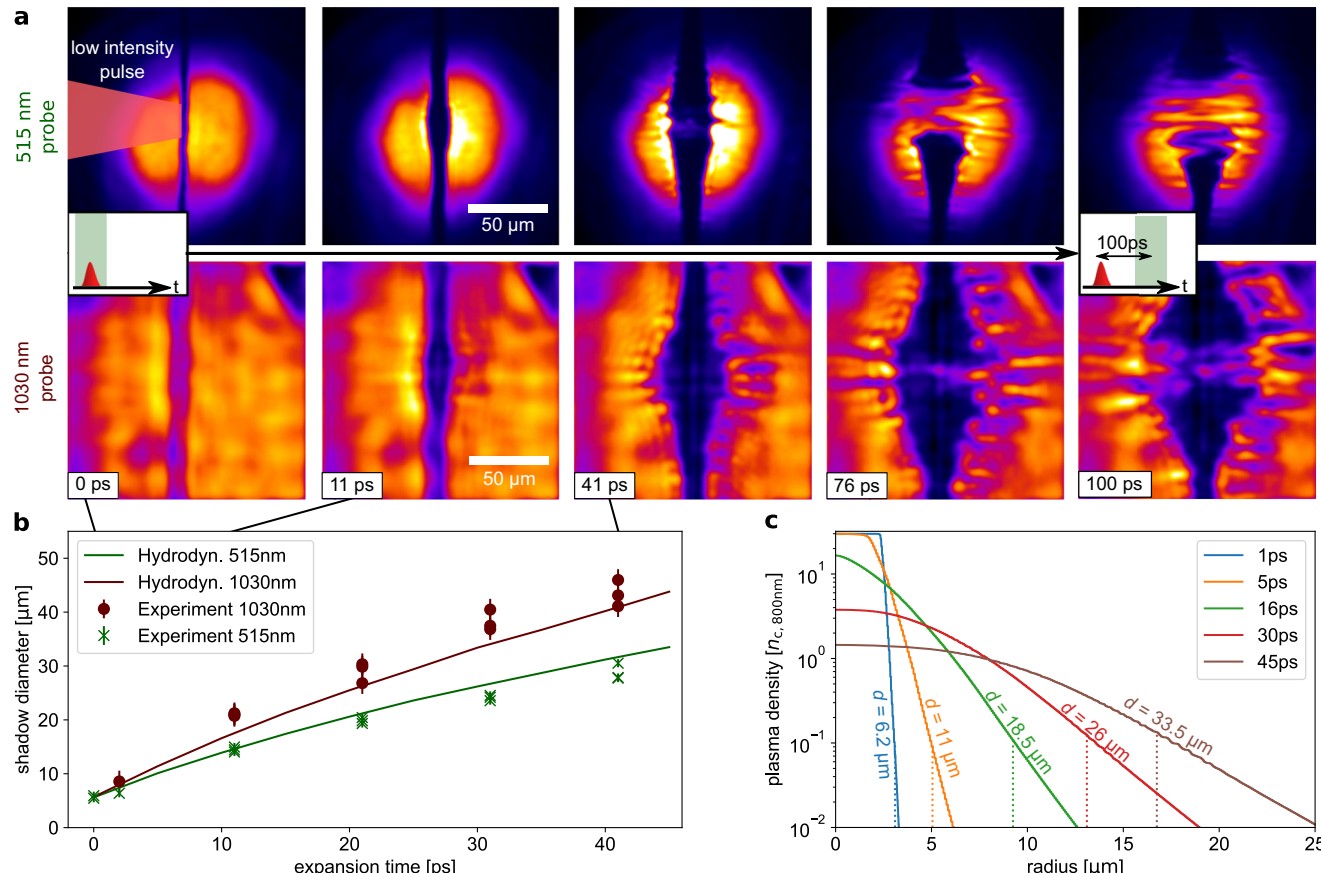

**Fig. 4 | Target expansion study and hydrodynamic simulation. a** Evolution of the jet target for different expansion times (given in the lower left corner) recorded using two-color optical shadowgraphy. **b** Extracted shadow diameters as a function of the expansion time for 515 nm and 1030 nm probe illumination. Error bars indicate the uncertainties of the shadow diameter measurement. A hydrodynamic simulation with an initial plasma temperature of 150 eV yields the best agreement with respect to the measured target expansion. This is indicated by the solid lines showing the evolution of the expected shadow diameter obtained from simulated density profiles using ray tracing. **c** Plasma density profiles for different expansion times according to the hydrodynamic simulation. The labels and dashed vertical lines indicate the shadow diameter as it would appear when probing each density profile.

the displayed data is still dominated by the spatial overlap quality and thus consider the best shots (orange dots in Fig. 3) for the discussion of the physical trends. A two-fold increase in energies to up to 80 MeV is observed between $d = 11 - 17\,\mu m$ with respect to the unexpanded jet. This trend is corroborated by more than 30 shots in which proton energies above the level of the best shot with an unexpanded target (38 MeV) were detected. At substantially larger diameters $d > 22\,\mu m$, energies remain above 20 MeV. A change in the proton emission distribution can be inferred from Fig. 3b showing the ratio of maximum proton energies $E_{max}$ recorded in 0° and in 45° directions. The best performing shots (orange dots) show an increase with shadow diameter which is supported by the binned averages (blue dashed line). No pre-expansion ($d = 5\,\mu m$) yields a ratio close to unity which is consistent with isotropic proton emission within the horizontal plane, i.e. perpendicular to the jet axis. This is expected for the curved surfaces of cylindrically shaped jets[41] and wire targets[51]. With increasing diameter, the emission direction of the most energetic protons is shifted towards the laser forward direction. This observation is qualitatively supported by proton profile measurements using a scintillator-based depth dose detector (see Supplementary Fig. 1). As a further signature, the pre-expansion of the target is associated with an increase of target transparency. This is indicated by the increasing fraction of light transmitted through the target with growing shadow diameter in Fig. 3c.

The pre-pulse tailored target density profile was determined in an independent experiment using high-resolution optical probing at two different wavelengths $\lambda$, addressing different critical densities as $n_c \propto \lambda^{-2}$. Time-dependent shadowgraphy images of the expanding target were simultaneously captured (see Fig. 4a) at wavelengths of 515 nm and 1030 nm and at a relative angle between the probes of 67°. Both series of images show an overall symmetric target expansion. Deviations from the idealized shape, e.g. axial density modulation (visible at expansion times starting from 40 ps and located within the under-dense plasma corona), add a small uncertainty to the specific shadow diameter determination. A larger shadow diameter is measured with the 1030 nm beam and at later expansion times, the entire bulk becomes transparent to the probe light, happening earlier for the 515 nm probe wavelength (at 41 ps versus 76 ps for 1030 nm probe wavelength). These observations indicate a radially symmetric plasma expansion with a surface density gradient. To infer the radial density profile for a given measured shadow diameter, one-dimensional hydrodynamic simulations were conducted. Assuming an initially homogeneous plasma at a certain temperature induced by the low-intensity pulse, the temporal density profile evolution is simulated. The connection between the simulated density profile and measured shadow diameter was established using a ray-tracing model of the probe imaging system (see methods). The shadow diameter as it would appear when probing a given density profile (c.f. vertical lines in Fig. 4c) was calculated for each time step of the simulation. The comparison of the temporal evolution of the measured shadow diameters and the simulated diameters for different initial temperatures yields the best agreement for a temperature value of 150 eV as shown in Fig. 4b. Based on this expansion study, it is possible to derive and confirm the density profile in the full-scale experiment by the on-shot

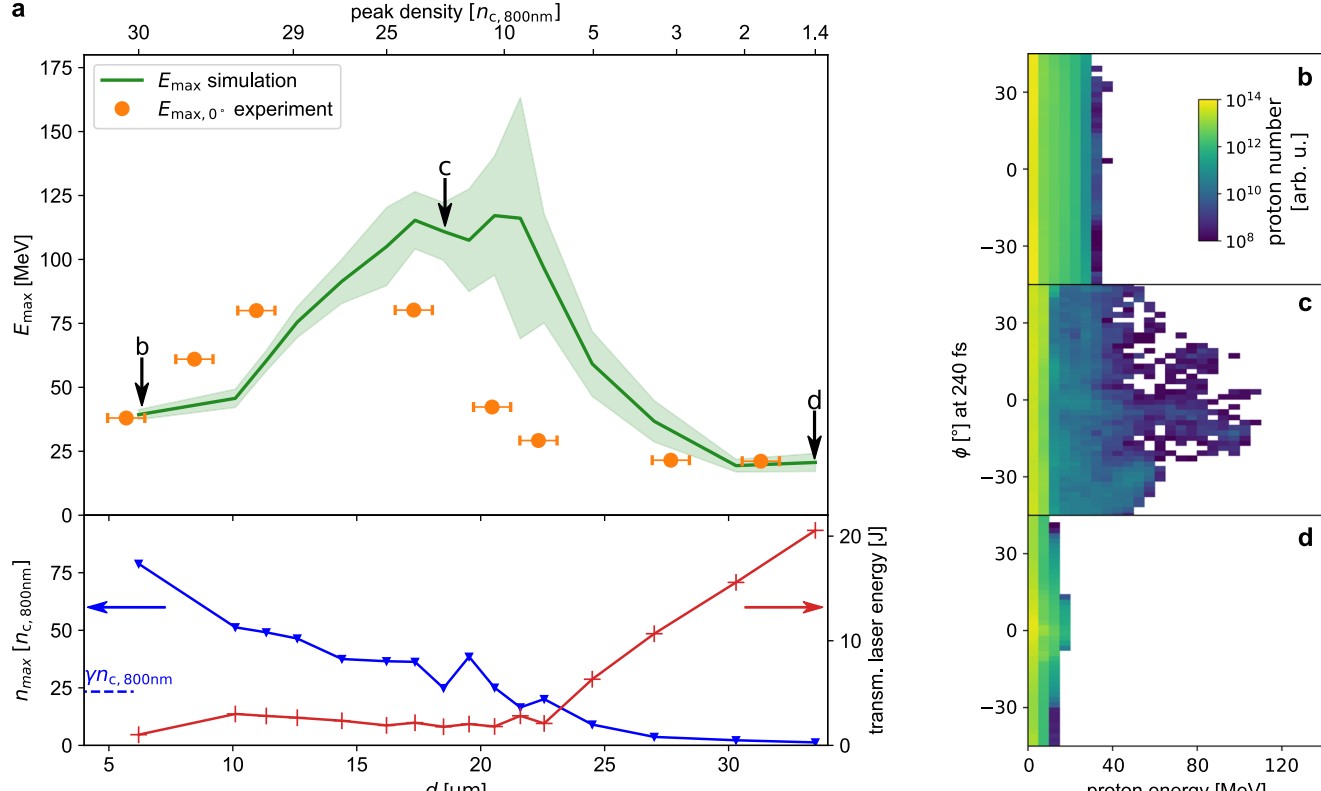

**Fig. 5 | Simulation results of the density scan. a** Summary of the conducted simulation runs using a normalized laser amplitude of $a_0 = 33$. The plasma density profile is represented by the shadow diameter $d$ (lower x-axis) and the peak density at the target center (upper x-axis). Maximum proton energies $E_{max}$ calculated at 240 fs are displayed in the upper part. 0 fs refers to the time when the peak intensity reaches the target center. The green line shows the energy for each target configuration averaged over 9 simulation runs and the shaded green area represents the standard deviation (refer to Supplementary Fig. 5). For simulation runs with average $E_{max}$, the maximum on-axis density $n_{max}$ (blue triangle, retrieved at 0 fs) are shown below. The blue dashed line indicates the theoretical threshold for relativistic transparency $\gamma n_{c,800nm}$. The bottom part also shows the laser pulse energy transmitted through the target (red plus). For comparison to the experiment, proton energies of the best-performing shots (orange circles as in Fig. 3a) are included as well. Error bars indicate the uncertainties of the shadow diameter measurement. **b–d** display the calculated (at 240 fs) proton energy spectra in the xy-plane as a function of the emission angle $\phi$ for representative simulation runs indicated by arrows in **a** with $d = 6.2\,\mu m$ (**b**), $d = 18.5\,\mu m$ (**c**) and $d = 33.5\,\mu m$ (**d**). $\phi = 0°$ indicates the laser propagation direction.

measurement of the shadow diameter with only one probe wavelength. The shots with the highest proton energies correspond to expanded target profiles (between orange and green profiles in Fig. 4c) with a moderate reduction of the density in the center (peak density) from 30 $n_{c,800nm}$ to 21 $n_{c,800nm}$ and still sharp density gradients whereby their existence constitutes an essential feature of the hydrogen jet target concept.

Summarizing, together with the measured increase in target transparency, the modification of the angular proton emission distribution, and the significant gain in proton energies, this finding indicates a change of the dominant acceleration mechanism in the transition to relativistic transparency.

**Simulating the density scan**

Based on the quantitative representation of the pre-pulse tailored density profiles derived from the hydrodynamic simulations and realistic assumptions concerning the temporal laser pulse rising edge[16], three-dimensional simulations were performed using the particle-in-cell code PIConGPU to identify the dominant acceleration mechanisms we walked through during the experiment (for simulation parameters see methods). Figure 5 summarizes the main results by presenting maximum proton energies, retrieved maximum on-axis plasma densities ($n_{max}$) and transmitted laser energies as functions of the target expansion state represented by the shadow diameter and the peak density of the plasma profile. Small diameter ($d < 11\,\mu m$) and a high peak density yield moderate proton energies of around 40 MeV.

Significantly higher $E_{max}$ exceeding 100 MeV are observed in the simulations for diameters around 20 $\mu m$. Within this region the maximum on-axis density drops below the theoretical threshold for relativistic transparency $\gamma n_{c,800nm}$ (blue dashed line) with $\gamma = \sqrt{1 + a_0^2/2}^2$ and $a_0$ the normalized laser amplitude. For larger diameters, laser energy transmission increases and maximum proton energies decrease. As the most prominent result of this study, target pre-expansion yields significantly higher maximum proton energies than the unexpanded case, a trend being in good agreement with the best-performing shots of the experiment (orange dots in Fig. 5a). For relativistically underdense initial target conditions, simulation and experiment yield lower proton energies and increased target transparency. For the limits of the unexpanded and the largely expanded case, the simulations are in near-perfect quantitative agreement with the experiment confirming the correct modeling of the energy transfer from the laser to the plasma. The visible offset of the simulation curve with respect to the shadow diameter, we ascribe to model-related limitations, such as the assumption of a thermal electron population for the pre-pulse induced target pre-expansion. Thus, the optimum of the simulation would be shifted to smaller shadow diameters for non-thermal contributions or smaller initial target diameters (see methods). Good agreement between experiment and simulation could also be found for the transition of the planar isotropic emission of the most energetic protons to a more laser-forward-directed emission with decreasing initial target density as illustrated by angular emission distributions from three sample simulations in Fig. 5b–d. Thereby, the

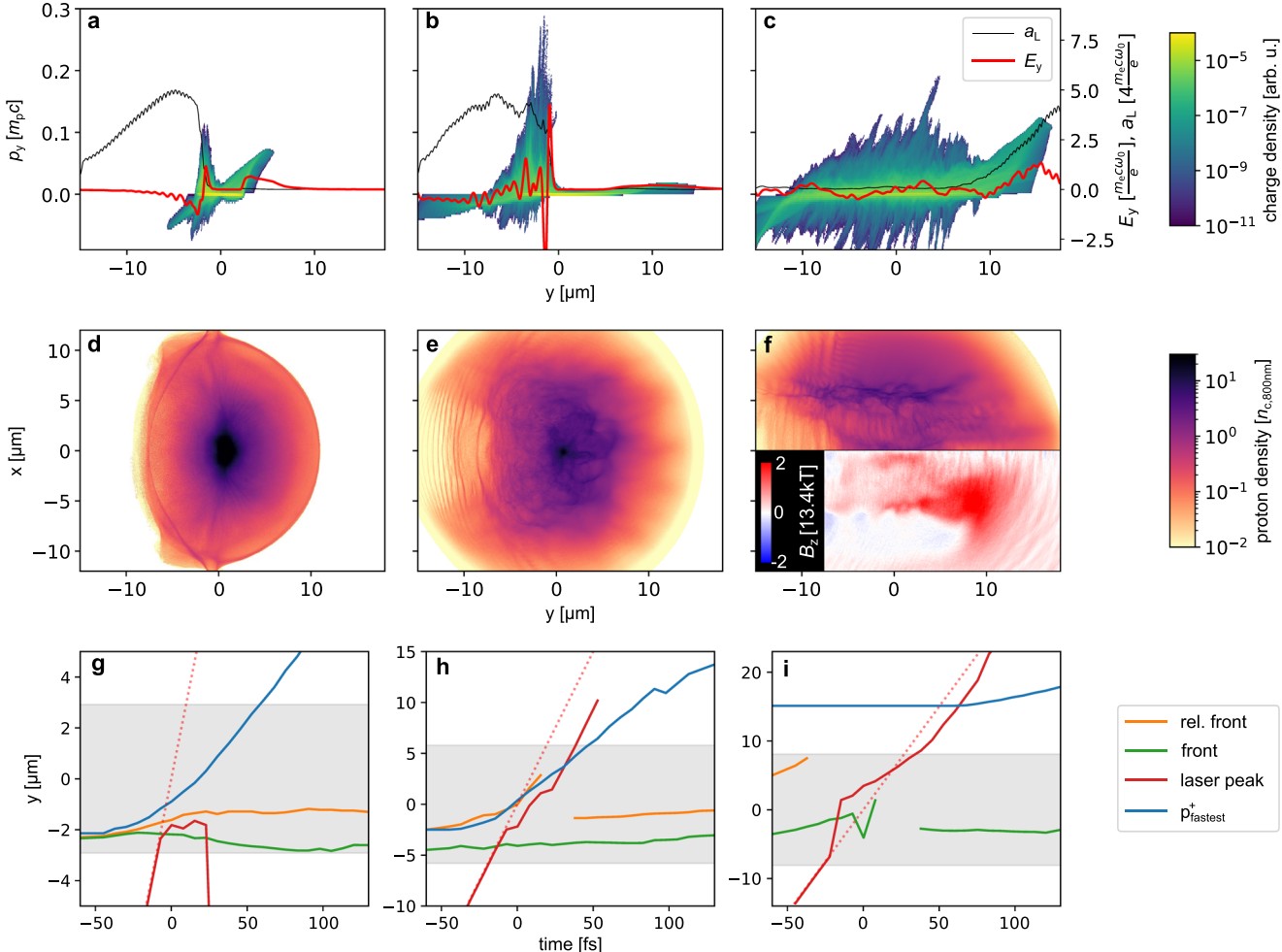

**Fig. 6 | Simulation results illustrating the different acceleration regimes.** For representative simulation runs with $d = 6.2\,\mu m$ (left column), $d = 18.5\,\mu m$ (middle column) and $d = 33.5\,\mu m$ (right column) longitudinal proton phase spaces (**a**–**c**) and proton density distributions (**d**–**f**) are presented. The corresponding target density profiles are displayed in Fig. 4c. The first row shows the longitudinal proton phase spaces either at -22 fs (**a**, **b**) or at 75 fs (**c**). On-axis electric field component ($E_y$) and local laser strength parameter ($a_L = [2I_L/(n_{c,800nm}m_ec^3)]^{1/2}$ with $I_L$ the laser intensity) are overlaid to demonstrate the accelerating field structure and the strength of the pulse, respectively. In addition to the proton density distribution (**d**–**f**) in the xy-plane at +120 fs, the $B_z$ component is shown in the bottom half of **f** to demonstrate the magnetic field structure. The third row (**g**–**i**) provides the temporal evolution of the position of the front at which the plasma density reaches the critical (green) as well as the relativistically critical density (orange). Additionally, the position of the laser pulse peak (red) and of the highest energy protons (blue) is shown. The grey shaded area in the third row indicates the region in which the target density is initially above $n_{c,800nm}$. The red dotted line shows the theoretical laser peak position in the case when no target would be present. Note the different y-scales in **g**, **h** and **i**.

emission direction and energy of the most energetic particles sensitively vary with the exact target position, which is demonstrated by dedicated simulation runs of non-perfect laser target overlap (see Supplementary Discussion). Together with the limited acceptance angle of the TPS, this study confirms our hypothesis that the proton energy fluctuation in the experiment is dominated by the spatial laser target overlap for all density regimes. In the optimal expansion case, an additional instability of the physics mechanism (as explained below) as well as an increased sensitivity to the laser input parameters occurs. In combination, these circumstances suggest why the maximum energies predicted by the simulations were not perfectly matched in the experiment.

The microscopic interaction process of three representative sample simulation runs is discussed below using longitudinal proton phase spaces and distributions of ion densities shown in the first and second row of Fig. 6, respectively. The third row provides the temporal evolution of the position of the critical (green) as well as the relativistically critical density front (orange), of the highest energy protons (blue) and of the peak of the laser pulse (red).

Isotropic proton emission in the TNSA-dominated regime: The initial density ($30\,n_{c,800nm}$) exceeds the theoretical threshold for relativistic transparency $\gamma n_{c,800nm} = 23.4\,n_{c,800nm}$. Additional steepening of the plasma interface induced by the compression of the electron fluid due to the radiation pressure further increases the maximum on-axis density (Fig. 5a). The target thus remains opaque, as confirmed by the non-penetrating laser pulse in Fig. 6a and the position of the laser peak in Fig. 6g. Fast electrons generated at the irradiated interface build up a quasi-static electric sheath field around the target, which is illustrated by the $E_y$ component in Fig. 6a, accelerating protons in the TNSA regime. Due to the curved surface, the protons (Fig. 5b) are emitted isotropically in the xy-plane. In parallel to the TNSA momentum spike at the target rear surface, a second spike of forward accelerated protons emerges at the target front surface in the proton phase space diagram (Fig. 6a). This population is often observed in numerical studies[36,41] and stems from HB-RPA. These protons can subsequently be injected into the rear side fields and get post-accelerated[21,22]. While in the case depicted in Fig. 6a, d, g both contributions are similarly relevant, the front side acceleration begins to dominate with decreasing target density.

Optimized acceleration at the relativistic transparency front (RTF): As a representative example for optimal performance, the second column of Fig. 6 shows the results for $d = 18.5\,\mu m$. We observe that in contrast to the overdense case (first column) where the laser is reflected at the critical density surface of the target front, and to the underdense case (last column) where the laser penetrates the transparent plasma, the laser is reflected inside the target bulk at a dynamically moving step of the relativistically corrected electron density. At the location of the laser reflection the electric field $E_y$ exceeds the rear side TNSA fields by several times (Fig. 6h). The laser penetrates the target while it progressively turns the plasma partially transparent by virtue of relativistic mass increase of the electrons oscillating in the intense laser field. The relativistic transparency front (RTF) that separates the relativistically transparent region from the opaque plasma with density above the relativistic critical density, moves forward, much slower than the vacuum speed of light (orange line in Fig. 6h). This allows the ponderomotive charge separation field that accelerates the protons, to follow protons into the bulk. As the laser intensity increases, the velocity of the RTF increases, which allows the protons to experience a sustained acceleration (Fig. 6h). This process stops as soon as the laser reaches the density downramp where the plasma becomes relativistically transparent (cp. Fig. 6h at 22 fs). The details of this mechanisms were described in[28,29]. Another interesting observation in the simulations is their instable nature specifically in the near-critical density regime. While for the unexpanded and largely expanded cases running the simulation with the same input parameters several times yield almost the same maximum proton energies, here we find large variations (green shaded area in Fig. 5). Similar variations in PIC simulation were observed in[52] and we verified the existence and magnitude of these fluctuations in the present near critical region to be largely independent from the PIC resolution and macro particle number in an extensive one- and two-dimensional parameter scan. The type of mechanisms at the RTF explains this observation. The synchronization of the proton acceleration is sensitive to the laser propagation inside the plasma. The laser propagation, however, depends on the local plasma heating, which varies due to initial small fluctuations in phase space that quickly grow during the laser propagation.

Magnetic vortex acceleration: For a relativistically underdense target at a diameter of $33.5\,\mu m$, the laser pulse penetrates the entire volume, as illustrated by the channel formation in Fig. 6f, the increased transmitted laser energy in Fig. 5a and the position of the laser maximum in Fig. 6c. The strong electric field at the laser front now moves too fast to efficiently accelerate protons in the bulk which is indicated by the broad momentum distribution behind the laser front in Fig. 6c. Efficient acceleration of protons is present at the target rear side (see Fig. 6c) due to the magnetic vortex acceleration (MVA) mechanism[31,32]. Inside the channel the laser generates a large electron current density in a non-linear wake-field which, when exiting the plasma, causes the formation of a toroidal magnetic field $B_z$ with a magnitude of several 10 kT (see bottom half of Fig. 6f). This induces an accelerating, transversely confined electric field, which together with the electrostatic field between the charged rear surface and the electron bunch leads to efficient proton acceleration along the laser propagation direction (see Fig. 5d).

## Discussion

High energy proton beams with up to 80 MeV were demonstrated in a proof-of-concept experiment employing a high-repetition-rate, debris-free cryogenic hydrogen jet target irradiated with ultra-short laser pulses in the PW regime. Low-intensity prepulse-induced hydrodynamic expansion was applied to tailor the target density to tune advanced acceleration schemes with a single target concept. In comparison to the unexpanded case, a two-fold increase of proton energy was achieved for the best shots reaching optimal pre-expansion to near-critical density. Remaining proton energy fluctuations could almost entirely be associated with the limited laser target overlap that can in principle be mitigated to a large extent by the use of wider sheet-like jets[41,47] in future studies. Based on hydrodynamic simulations and two-color optical probing of the prepulse-induced target pre-expansion, a target density profile model was developed. This allowed for comparison of experimental signatures with results of realistic three-dimensional particle in cell simulations. While the quantitative agreement between experiment and simulation in the overdense and the underdense regimes was reached, energy enhancement ratio, emission characteristic and the prediction of the optimal near-critical target density range were found to be remarkably consistent. In the near-critical regime enhanced proton acceleration at the relativistic transparency front progressively penetrating into the target bulk can be concluded from the simulation. Further studies may achieve more favorable ion beam parameters through fine-tuning of the target density profile, which was in retrospect not fully optimized. According to simulation, at more homogeneous density profiles and steeper gradients, energies exceeding 100 MeV are expected for matched laser and target conditions at current systems for both optimized RTF-RPA[29] and MVA[31] scenarios.

Without the use of the plasma mirror device representing otherwise the main limitation of real repetition rate operation, we also achieved single shot energy performance of 80 MeV, yet under uncontrollable pre-plasma conditions (see Supplementary Fig. 6). Looking beyond the conceptional breakthrough, this study demonstrates that the combination of cryogenic jet targets and PW-class laser provides a path to developing 100 MeV-class, high repetition rate proton accelerators based on existing laser technology.

## Methods

### Experimental setup

The experiment was performed using the Titanium:Sapphire based laser system DRACO PW[53] delivering linearly polarized pulses with 18 J energy on target and a duration of 30 fs yielding peak intensities of about $5.4 \cdot 10^{21}\,W\,cm^{-2}$ ($a_0 = 50$) at a focal spot size of $2.6\,\mu m$ (FWHM). A recollimating single plasma mirror device[16] was applied for temporal contrast cleaning. A solid hydrogen jet target[47,54] with an electron density of $5.1 \cdot 10^{22}\,cm^{-3} = 30\,n_{c,800nm}$ was produced by injecting pure liquid hydrogen at a temperature of about 18 K through an aperture of 5 μm diameter into vacuum where evaporative cooling caused solidification. With a velocity of around 100 m s$^{-1}$, the jet is continuosly refreshing allowing for high repetition rate experiments[55]. By implementing a mechanical chopper blade, the target system could be protected from the disruptive laser-plasma interaction and thus the laser focus could be aligned 10 mm below the nozzle.

Target pre-expansion was induced by low-intensity prepulses generated from a pick-off mirror (one inch in diameter) inserted in front of the last folding mirror before final focusing. A variable distance between the pick-off and beamline mirror allowed controlling the delay between the main pulse and the prepulse. Light was reflected at the rear surface of the pick-off mirror increasing the prepulse duration to 55 fs due to the additionally induced group delay dispersion in the 6 mm thick mirror substrate. The prepulse was focused using the same final focusing optic as for the main pulse resulting in a spot size of 32 μm × 19 μm (FWHM) and thus a peak intensity of $5.8 \cdot 10^{17}\,W\,cm^{-2}$.

### Diagnostics

Proton energy spectra were recorded by two Thomson parabola spectrometers (TPS), implemented at 0° (main laser propagation direction) and at 45° (see Fig. 1). Each TPS was equipped with a microchannel plate (MCP) containing a phosphor screen that was imaged onto a camera for on-line readout. The lower energy cut-off of

the TPS is 4 MeV and the energy resolution $\Delta E$ dominated by the projected size of the pinhole onto the MCP amounts to $\Delta E = \pm 2$ % for 10 MeV protons and increases to $\Delta E = \pm 4.5$ % at 80 MeV. The energy calibration of the TPS (930 mT magnetic field strength over a nominal length of 200 mm) was conducted at a clinical proton acceleration with an energy and charge-calibrated beam. This serves as a benchmark for the relativistic particle tracing simulation that provide the input for processing the TPS images. As only protons are accelerated, no overlapping traces from other ions exist that would otherwise complicate the analysis of the proton spectra.

Stacks of calibrated radiocromic films (RCF) covering a solid angle of 0.17 sr were occasionally inserted behind the target (0° axis) to measure the proton beam profile and to cross-calibrate the particle yield measured with the TPS spectrometers.

A screen of glass ceramic was installed at a distance of 450 mm behind the laser focus to record the residual transmitted laser light. The screen was imaged onto a camera equipped with a band-pass filter (800 ± 20 nm). The captured light signal consists of two components: laser light passing around the target and laser light transmitted through the target. Assuming central hits, the first contribution can be estimated and subtracted by calculating the corresponding fraction of the light passing around the target. This is accomplished by blocking the region of the expanded or unexpanded jet in the focal plane which is deduced from the measured probe image of each shot. We note that, especially for strongly expanded targets, not all the transmitted light is collected due to diffraction to outside the ceramic screen boundaries resulting in an underestimation of the transmission values.

High-resolution optical shadowgraphy imaging via two long working distance microscope objectives was implemented along 69° and 134° with respect to the laser propagation direction. Back-illumination of the target was accomplished by two different probe wavelengths of 515 nm and 1030 nm, respectively (details are reported in[48]). The latter was primarily applied for characterization of the target expansion when using low-intensity pulses ($2.7 \cdot 10^{18}$ W cm$^{-2}$ 30 fs, 250 mJ). The resolution limit of the imaging systems was measured to be 1.5 $\mu$m for the 515 nm probe arm and 4 $\mu$m for the 1030 nm probe arm. The probe beams were generated with a stand-alone laser system delivering pulses with an energy of 1 mJ and a duration of 160 fs[56]. The oscillators of DRACO and the probe laser system were temporally synchronized with an optoelectronic system. For every shot the probe delay was precisely determined with a resolution of 175 fs using an auxiliary beam arrival monitor system[48].

### Identification of central hits

Since the measurement of the on-shot jet position and the high power focus position was not sufficiently precise, the degree of laser target overlap was determined by integrating the laser light signal measured with a ceramic screen behind the target (cf.[41]). For a central interaction, the target blocks a large part of the focus spot leading to reduced laser light signal on the ceramic screen, whereas more light is recorded on the screen if the laser pulse only grazes along the jet surface or misses the target completely (see Supplementary Fig. 7b). For course identification of central hits (blue dots in Fig. 3), we applied a threshold selecting one-third of all shots with the lowest laser light signal of a given subset, that contains same sized targets. (Supplementary Fig. 7a shows all data points.) Although the fraction of 1/3 was obtained empirically, the trend of the average maximum proton energies remains robust against changing this fraction. Note that the contribution of transmitted laser light (see Fig. 3c) to the total signal measured on the ceramic screen is small and thus the identification of central hits remains unaffected by this.

### Particle in cell simulations

The three-dimensional Particle in cell (PIC) simulations were performed using the fully relativistic code PIConGPU[57,58] version 0.4.3. The

target model has cylinder symmetry and translational symmetry along the z-axis. In the radial direction the density profiles derived from the hydrodynamic simulations was used (see Fig. 4c). A total of 16 different profiles with corresponding shadow diameters between 6.2 $\mu$m and 33.5 $\mu$m were simulated. The laser pulse is linearly polarized in the x-direction, propagates in the positive y-direction and has a wavelength of 0.8 $\mu$m. The laser pulse has Gaussian shape in transverse direction yielding a spot size of 4.9 $\mu$m (FWHM) and a normalized peak vector potential of $a_0 = 33$ that is slightly smaller than the peak intensity in the experiment to cope with the imperfections during the measurements (e.g. systematic alignment differences between the jet position along the laser axis and the focal plane, day-to-day alignment differences, shot-to-shot position fluctuations). We modeled the temporal laser distribution using a gaussian profile with 30 fs (FWHM) pulse duration for the peak intensities and an exponential ramp with parameters fitted to the experimental measurements[16]. The ramp is defined by an intensity ratio compared to the peak intensity of $5.55 \cdot 10^{-8}$ at −600 fs and 0.1894 at the peak time. To ensure energy conservation, the energy for this ramp is taken out of the main pulse. The simulation runs for 240 fs after the pulse peak reaches the target center. The simulation box size is 14.4 $\mu$m in z-direction and 28.8 $\mu$m in x-direction. In the longitudinal direction the target center is placed 14.4 $\mu$m from the box border on which the laser enters and the box extends for another 43.2 $\mu$m on the rear side. The laser wavelength is resolved by 24 cells in every direction. The timestep fulfills $\Delta t = \Delta x / (\sqrt{3} \cdot c) = 0.064$ fs. The target material of ionized hydrogen is modelled with a varying number of macroparticles per cell, depending on the pre-expansion, of third-order cloud shape. For the smallest target 18 particles per cell were used. Due to memory limitations with the large plasma-filled volume only 3 particles per cell could be used for the largest target configuration. The PIC cycle uses the Yee field solver, Esirkepov current deposition and Boris particle push. Boundary conditions are treated as absorbing in all directions.

The transmitted laser energy (see Fig. 5a) is determined by integrating the energy density of the electromagnetic field for y > 20 $\mu$m at a simulation time of +120 fs. At this time, the majority of the remaining laser pulse is positioned at larger y-coordinates but still inside the simulation box. Contributions from the target matter and associated fields are predominantly located at smaller y-coordinates and are therefore excluded. Similar to the experiment, laser energy passing around the target is subtracted. This contribution is estimated by integrating the laser fluence in the focal plane over the transparent region in which the target density is lower than $n_{c,800nm}$.

The position of the relativistic transparent front in Fig. 6g−i indicates the lowest y-value where $n_e$ increases above $0.7 \gamma n_{c,800nm}$. No position is shown if $n_e$ is below this density threshold inside the entire target. The position of the (non-relativistic) density front is the lowest y-value where $n_e$ increases above $0.7 n_{c,800nm}$.

### Ray-tracing

Synthetic shadowgrams were calculated for each plasma density distribution using the ZEMAX software package. This ray-tracing approach models the refraction in the plasma density distribution as well as the propagation through the imaging system to the detector plane. The simulated shadow diameter is then determined by the size of the shadow in the synthetic detector images. Note that the density at this diameter is more than one order of magnitude below the critical density (slightly dependent on the plasma profile) for the corresponding probe wavelength[48].

### Hydrodynamic simulations

Hydrodynamic simulations were conducted using the 1D-FLASH code[59,60] (assuming radial symmetry) with the equation of state for hydrogen. The target is modeled using a constant density of 30 $n_{c,800nm}$ spanning over a radius of 2.5 $\mu$m. The interaction of the

low-intensity pre-pulse is assumed to heat the target isochorically, leading to a homogeneous plasma at a certain temperature. We justified this assumption using 2D PIC simulations that indicate thermalization after a few 100 fs for the same conditions. We further conducted cylindrically symmetric 2D-FLASH simulations to study the temperature gradient along the jet axis, which stems from the intensity distribution from the pre-pulse. The influence of corresponding heat transfer on the expansion dynamics can be neglected. By comparing the simulated diameters obtained by ray-tracing with the experimentally measured shadow diameters, the best matching temperature (150 eV) was determined. Higher or lower temperatures, that are e.g. plausible within the uncertainties of the experiment (100 eV to 250 eV), primarily lead to faster or slower expansions, while the overall shape of the plasma density profile at a certain expansion state remains similar. Hence, the relationship between the shadow diameter and the plasma density profile is independent of the initial temperature estimate. The limitation of our procedure to transform the optical probe images into a plasma density profile can result in a systematic shift of the peak density with respect to the optically observed shadow diameter. One limitation is that our method inherently assumes a thermal electron population. However, a non-thermal plasma could lead to long low-density corona plasma and by that would shift the peak density (and by that the simulation results illustrated by the green line in Fig. 5) to the left. More complex density profiles would complicate the assignment between the simulated density profile and the measured shadow diameter. Another yet less important source of error is an inaccurately determined initial jet diameter, where FLASH simulations show a respective fast reduction of the peak density for smaller shadow diameters (i.e. causing again a shift of the simulation data to the left). This, however, can only account for about 20% of the shift in peak density.

## Data availability

The data that support the findings of this study is available via Rodare at https://rodare.hzdr.de/record/2311 (ref. 61).

## Code availability

Code written for the use of this study is available upon reasonable request from the corresponding author.

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

## Acknowledgements
We would like to acknowledge the DRACO laser team for their excellent support. The work of C.B.C., F.F., M.G., S.H.G., J.B.K., F.T. and C.S was supported by the U.S. Department of Energy, Office of Science, Fusion Energy Science under FWP 100182. C.B.C. acknowledges partial support from the Natural Sciences and Engineering Research Council of Canada (NSERC). F.T. acknowledges support from the National Nuclear Security Administration (NNSA). This work was partially funded by the Center of Advanced Systems Understanding (CASUS) which is financed by Germany's Federal Ministry of Education and Research (BMBF) and by the Saxon Ministry for Science, Culture and Tourism (SMWK) with tax funds on the basis of the budget approved by the Saxon State Parliament. The software used in this work was developed in part by the DOE NNSA- and DOE Office of Science-supported Flash Center for Computational Science at the University of Chicago and the University of Rochester.

## Author contributions
M.Reh, S.A., C.B., F.-E.B., C.B.C., L.G., M.Gau, S.G., J.B.K., S.K., F.K., J.M.-N., M.L., L.O.-H., M.Rei, H.-P.S., C.S., M.S., F.T., T.Z. and K.Z. executed the experiments and contributed to diagnostics development. M.Reh, C.B. and K.Z. analyzed and interpreted the experimental results. I.G., M.Gar, A.H., T.M. and T.K. performed numerical PIC simulations. T.E.C, L.H. and L.Y. conducted the hydrodynamic studies. M.B., T.E.C, F.F, S.H.G, U.S. and K.Z. supervised the project. All authors contributed to discussions and revision of the manuscript.

## Funding

## Competing interests
The authors declare no competing interests.
