## [Peer Review File · Nature Communications]

Ultra-short pulse laser acceleration of protons to 80 MeV from cryogenic hydrogen jets tailored to near-critical densityEditorial Note: This manuscript has been previously reviewed at another journal that is not operating a transparent peer review scheme. This document only contains reviewer comments and rebuttal letters for versions considered at *Nature Communications*.

REVIEWER COMMENTS

Reviewer #1 (Remarks to the Author):

After reading the authors' reply to my 2nd round comments, I remain horrified when reading "relativistic mass increase" but unfortunately this is not sufficient to reject the manuscript. After having also read the reply to the other reviewer, I think that the revised manuscript can be published in Nat. Comm. as it stands.

Reviewer #4 (Remarks to the Author):

I have read the manuscript together with the comments from the two previous reviewers. The manuscript reports on a platform for accelerating protons with high power lasers which is overall novel and very interesting. This is based on the controllable expansion of a cryogenic hydrogen jets, combined to an ultrashort PW laser.

The authors show the possibility of changing the interaction and proton acceleration regimes, by varying the time interval between a heating prepulse and the main interaction pulse. Besides an extensive characterization of the proton beam, the key parameter for characterizing the interaction conditions is the shadow of the dense plasma, which is measured by dual wavelength optical shadowgraphy and compared to computational models of the plasma expansion and probe ray tracing.

The data are very interesting, the analysis and interpretation is detailed and convincing, and of high quality throughout.

Referring to the criticisms of previous reviewers (and in particular of Reviewer #3, who was the most critical), I agree that there have been already several demonstrations of proton acceleration in relativistic transparency regimes, and also that acceleration from cryogenic targets has been demonstrated in several experiments.

However, while the experiment reported here builds on some of this elements, it presents a novel combination of different aspects which makes the overall novelty of the work significant and of high interest to researchers in the field. Furthermore, although the energies demonstrated are not a record

in absolute terms, I understand that 80 MeV energies are arguably the highest energies reported for ultrashort PW interactions (if one excludes results which are broadly recognised to be spurious).

Highest energies, approaching 100 MeV, have been reported on the VULCAN PW laser, where however 500 J of energy are contained in much longer pulses (0.5-1 ps), i.e. interaction conditions which are significantly different from the ones reported here.

Some other regimes reported by referee 3 are not directly relevant to the conditions of this paper: optimization of the preplasma conditions as reported by Gizzi et al, Sci. Rep. 11, 13728, 2021 is relevant to TNSA acceleration from relatively thick foils, but not to the conditions reported by the author. Magnetic Vortex Acceleration is similarly not relevant, therefore any previous results in these areas do not undermine the novelty of this paper.

Operation at high-repetition rate is also not the main focus of this paper, so criticising the current unsuitability of this approach to high-rep operation is fair to some extent, but I do not see this as a major obstacle to publications.

The authors are very honest and open about the unstable nature of the output, and their current inability to control the interaction conditions. In the paper it is stated that the main issue is the difficulty in overlapping perfectly the focal spot with the limited transverse extent of the plasma.

I have to say that, while I find the manuscript interesting and well written, and the arguments put forward are generally convincing, I am a little troubled by the results shown in Fig 3(a) and Fig.7(a)(supplementary file) and I would be interested in hearing the authors' view on some related queries, as discussed below. In these graphs it seems that the best results indicated by the orange points are quite detached from the rest of the results in the range of 10-20 micron diameters. I would have expected these to be the limits of a range of energy values, as determined by the variability in transverse overlap. There is instead a significant gap between these optimal energies and the energy values just below. Are these just single shot results? Can some statistical considerations be made on why these are not happening more often? And also, could the authors comment on why there is such a large energy gap between the best results and the second best energy values (almost 20 MeV in some cases)?

Dear reviewers, dear editor,

on behalf of all authors, we thank you for the effort you put into your reports and for acknowledging the quality of our manuscript and importance of our work. Please find our response to the reviewer's comments below in **BLUE**. We further made small changes to the supplementary information according to your guidance and comments. These changes are marked in **RED**.

Comments from the reviewers:

Reviewer #1 (Remarks to the Author):

After reading the authors' reply to my 2nd round comments, I remain horrified when reading "relativistic mass increase" but unfortunately this is not sufficient to reject the manuscript. After having also read the reply to the other reviewer, I think that the revised manuscript can be published in Nat. Comm. as it stands.

We thank the reviewer for his work in reviewing our manuscript and his recommendation for publication.

Reviewer #4 (Remarks to the Author):

I have read the manuscript together with the comments from the two previous reviewers. The manuscript reports on a platform for accelerating protons with high power lasers which is overall novel and very interesting. This is based on the controllable expansion of a cryogenic hydrogen jets, combined to an ultrashort PW laser.

The authors show the possibility of changing the interaction and proton acceleration regimes, by varying the time interval between a heating prepulse and the main interaction pulse. Besides an extensive characterization of the proton beam, the key parameter for characterizing the interaction conditions is the shadow of the dense plasma, which is measured by dual wavelength optical shadowgraphy and compared to computational models of the plasma expansion and probe ray tracing.

The data are very interesting, the analysis and interpretation is detailed and convincing, and of high quality throughout.

Referring to the criticisms of previous reviewers (and in particular of Reviewer #3, who was the most critical), I agree that there have been already several demonstrations of proton acceleration in relativistic transparency regimes, and also that acceleration from cryogenic targets has been demonstrated in several experiments.

However, while the experiment reported here builds on some of this elements, it presents a novel combination of different aspects which makes the overall novelty of the work significant and of high interest to researchers in the field. Furthermore, although the energies demonstrated are not a record in absolute terms, I understand that 80 MeV energies are arguably the highest energies reported for ultrashort PW interactions (if one excludes results which are broadly recognised to be spurious).

Highest energies, approaching 100 MeV, have been reported on the VULCAN PW laser, where however 500 J of energy are contained in much longer pulses (0.5-1 ps), i.e. interaction conditions which are significantly different from the ones reported here.

Some other regimes reported by referee 3 are not directly relevant to the conditions of this paper: optimization of the preplasma conditions as reported by Gizzi et al, Sci. Rep. 11, 13728, 2021 is

relevant to TNSA acceleration from relatively thick foils, but not to the conditions reported by the author. Magnetic Vortex Acceleration is similarly not relevant, therefore any previous results in these areas do not undermine the novelty of this paper.

Operation at high-repetition rate is also not the main focus of this paper, so criticising the current unsuitability of this approach to high-rep operation is fair to some extent, but I do not see this as a major obstacle to publications.

The authors are very honest and open about the unstable nature of the output, and their current inability to control the interaction conditions. In the paper it is stated that the main issue is the difficulty in overlapping perfectly the focal spot with the limited transverse extent of the plasma. I have to say that, while I find the manuscript interesting and well written, and the arguments put forward are generally convincing, I am a little troubled by the results shown in Fig 3(a) and Fig.7(a)(supplementary file) and I would be interested in hearing the authors' view on some related queries, as discussed below. In these graphs it seems that the best results indicated by the orange points are quite detached from the rest of the results in the range of 10-20 micron diameters. I would have expected these to be the limits of a range of energy values, as determined by the variability in transverse overlap. There is instead a significant gap between these optimal energies and the energy values just below. Are these just single shot results? Can some statistical considerations be made on why these are not happening more often? And also, could the authors comment on why there is such a large energy gap between the best results and the second best energy values (almost 20 MeV in some cases)?

We thank the reviewer for reviewing our manuscript, in particular in view of the previous reports, and for acknowledging the quality of our data and interpretation. We are grateful for the general confirmation of our case and the degree of novelty of our work, and for sharing our opinion that the work presented should be of significant interest to the community.

We agree that the limited reproducibility of the beam energy is the most important challenge that we have to tackle for the future application of our platform. The reasons behind the presented maximum energy distribution shown in Fig. 3a and Fig. 7a (supplementary material) have been studied intensely during data analysis and investigated by dedicated series of three-dimensional particle in cell simulation runs. The source of the rather large proton energy scattering in the target diameter range where the best shots are observed (10-18 μm) is due to a combination of three effects: the limited control over the geometric overlap of laser focus and the curved target, the highly non-linear nature of the acceleration schemes at the onset of transparency, and the limited acceptance angle of the Thomson parabola spectrometer. As these individual and conceptually different aspects are interconnected, we are not dealing with a typical statistical distribution of a data set but with a few top-performing single shots realized only under perfect conditions and thus representing the ideal physics case. The following arguments explain why we do not observe the shots with the highest energy more frequently.

First, the influence of deviations from perfect laser to target overlap (less than $2\mu\text{m}$) together with the limited acceptance angle of the TPS was discussed in the context of Fig. 4 of the supplement. As supported by the simulation studies, the maximum particle energy is very sensitive to the spatial offset and varies between 20 MeV and 80 MeV. As correctly assumed by the reviewer, this effect alone cannot explain the too low number of shots with energies above 60MeV (i.e. the energy gap below the best results) compared to shots with lower proton energies. (Please note that we do not observe such an energy gap in the case of the unexpanded jet because the non-perfect overlap represents the predominant contribution to the energy scattering in this case.)

For the target diameter range of the best shots, the nature of the acceleration schemes being sensitive to the synchronization of the acceleration field structure with the accelerated protons adds sensitivity to details of the laser input parameter (intensity, spatial-temporal couplings, temporal

contrast). In the simulations, this leads to the observed deviations between re-runs of the same input (see shaded green area in Fig. 5a of the manuscript) and, in the extreme case (Fig. 2 of the supplementary material) to large (factor 2) fluctuations of the proton energy in the laser forward direction. Already a reduction in the proton energy of 25% due to non-ideal laser parameters is sufficient to reduce the amount of the best performing shots.

In summary, we think that the highest energies are only reached in the case of a perfect laser target overlap as well as ideal on-shot laser parameters. High-energy shots are therefore less common than shots yielding lower energies (<60MeV). They occur when only one requirement is satisfied or both are only partially fulfilled.

To clarify the different causes of the energy scatter we have added a brief statement to the caption of the supplementary Fig. 7.

Thus, although we are confident that the technical (yet non-trivial) issues related to jet stability can be overcome by dedicated (e.g. flat surface) jet geometries and active stabilization in the future, full control over the synchronized particle acceleration at the onset of transparency requires further understanding of the processes at play as stated in the outlook.

REVIEWERS' COMMENTS

Reviewer #4 (Remarks to the Author):

I found the response of the authors to my queries regarding the source of severe proton energy fluctuations observed in the experiment, quite reasonable and convincing. This is clearly a significant limitation at present which will need to be addressed in view of any future application of the source. Nevertheless, I strongly support the paper for publication on Nature Communications. I reiterate my previous view that the results presented in the paper are novel and of broad interest, and that the analysis and modelling presented is convincing and of very high quality. The implications for future development of laser-driven proton sources are also important.

Dear reviewers, dear editor,

on behalf of all authors, we thank you for the effort you put into your reports. Please find our response to the reviewer's comments below in BLUE.

Comments from the reviewers:

Reviewer #4 (Remarks to the Author):

I found the response of the authors to my queries regarding the source of severe proton energy fluctuations observed in the experiment, quite reasonable and convincing. This is clearly a significant limitation at present which will need to be addressed in view of any future application of the source. Nevertheless, I strongly support the paper for publication on Nature Communications. I reiterate my previous view that the results presented in the paper are novel and of broad interest, and that the analysis and modelling presented is convincing and of very high quality. The implications for future development of laser-driven proton sources are also important.

We thank the reviewer for reviewing our manuscript, providing strong support for publication, and again highlighting the quality of our work as well as the novelty and importance of our results.